# Comparative Constructions in Zhoutun from a Language Contact Perspective

## Chenlei Zhou

Department of Syntax and Semantics, Institute of Linguistics, Chinese Academy of Social Sciences, Beijing 100732, China; zhoucl@cass.org.cn

**Abstract:** The paper describes comparative constructions in Zhoutun, a Chinese variety that was heavily influenced by Amdo Tibetan and spoken in Guide County, Qinghai Province. There are five comparative constructions (Cxn), based on the type of comparative marker, in Zhoutun, namely (1) the *xa*-Cxn; (2) the *pi*-Cxn; (3) the 'look'-Cxn; (4) the 'and'-Cxn; and (5) the hybrid Cxn. The five constructions illustrate features from both Chinese and Amdo Tibetan, and their co-existence demonstrates the mixed nature of the comparative constructions, as well as the grammar system of Zhoutun due to language contact. This paper also argues that the "comparative subject" should be further subcategorized into "comparative subject" and "attributive subject", and that the "comparative result" should be divided into "abstract measurement" and "concrete measurement" in the typological study of comparative constructions.

**Keywords:** comparative constructions; language contact; Zhoutun; Chinese; Amdo Tibetan

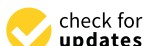



## 1. Introduction

Comparing two objects is a common mental act, and in typological study, the linguistically encoded constructions that express the comparison of inequality are known as comparative constructions (Stassen 2013). A typical comparative construction consists of four components: the comparative subject (CS), the standard (St), the comparative marker (CM), and the comparative result (CR), as shown in (1), where CS = John, St = Ben, CM = than, and CR = taller.

(1) John is taller than Ben.

The above-mentioned variables have been deemed to be the loci of interest in typological studies on comparative constructions. (See, among others, Dixon 2012). However, there are still some variables that merit further attention. This study provides evidence that the CS can be further sub-classified into comparative subject and attributive subject, and the CR can be further differentiated into concrete measurement and abstract measurement.

This paper investigates the comparative constructions in Zhoutun. Zhoutun is a Chinese dialect spoken by 800–900 native speakers living in Zhoutun Village, Guide County, Hainan Tibetan Autonomous Prefecture, Qinghai Province, P.R. China.

Zhoutun Village has a population of 882 residents, of whom 85% are Han, 10% are Tibetan, and 5% are Monguor/Tu. These demographic data are based on official statistics from 2014. The majority of the Han population, approximately 750 individuals, regularly use Zhoutun in their daily discourse. The current level of bilingualism among the Tibetan and Monguor/Tu communities in the village is not well documented and remains to be investigated. Based on my observations, the Monguor/Tu population tends to use Zhoutun when communicating with the Han population in the village, while the Tibetans predominantly use Tibetan. It is worth noting that the data collected in this study were limited to the Han population.

According to anecdotal evidence, some older Han individuals in Zhoutun Village, particularly those over 80 years of age, possess the ability to speak Amdo Tibetan. However, due to the limited number of such individuals and the prevalence of poor health among them, data collection from their interactions with Tibetans was unfeasible. Nevertheless, a personal informant, Qiulan Xu (born in 1933), attested that when she was young, the Han population in Zhoutun Village regularly engaged in communication with nearby Tibetans using Amdo Tibetan. The present study also identified a cohort of middle-aged Han speakers, roughly 40 years of age, who possess proficiency in Amdo Tibetan, particularly merchants who have frequent transactional interactions with Tibetan-speaking individuals in neighboring villages.

The younger generation in the Zhoutun Village demonstrates a reduced level of proficiency in the Amdo Tibetan language. This is attributed to two primary factors: the increasing social development in the area, which has led to a preference for relocating to more developed counties, and the mandatory education program that sends village teenagers to boarding schools in the county. These boarding schools foster an environment of increased inter-village communication, where the local variety of Mandarin known as Qiaohua (a variant of the Xining dialect spoken in Guide County, situated 22 km from the Zhoutun village) serves as the common language. As a result of these expanded communication channels, a significant proportion of older individuals in the Zhoutun village have also acquired proficiency in Qiaohua.

The influence of Mandarin Chinese on Zhoutun is apparent, primarily through the exposure to Mandarin-speaking media sources such as television programs. However, the impact of Mandarin is less pronounced compared to that of the regional variety Qiaohua. The utilization of Mandarin as a communicative tool with external speakers, such as the author, varies among the residents of the Zhoutun Village, according to their level of proficiency in the language. Additionally, the influence of Mandarin on Zhoutun is characterized by a marked interference in linguistic features, such as tone and word order.

Zhoutun is characterized by a hybrid analytic–agglutinative linguistic system, which exhibits a higher prevalence of clitics or affixes, such as postpositions and case markers. In terms of basic word order, Zhoutun displays a rigid SOV pattern, where the object always precedes the verb, regardless of whether it is definite or indefinite. This is in contrast to certain northwest Chinese varieties, such as Tangwang and Xining, which are spoken in the same region and exhibit mixed linguistic features. In these varieties, a definite object precedes the verb, while an indefinite object is positioned after the verb, adhering to the preferred SOV order. With regards to ditransitive constructions, the basic word order in Zhoutun is ARTV, where A represents the agent, R the recipient-like role, T the theme-like role, and V the verb, thereby conforming to the OV pattern.

At the subclausal level, including constituents such as Noun Phrases (NP) and Verb Phrases (VP), Zhoutun displays characteristics that are typical of OV languages.

(2)    OV correlation pairs found in Zhoutun:
    A. NP–adposition;
    B. Predicative–copula verb;
    C. Verb–tense/aspect auxiliary verb;
    D. Genitive–noun;
    E. Comparative standard–adjective;
    F. Prepositional phrase–verb;
    G. Manner adverb–verb.

The OV-pattern is without a doubt the result of prolonged and intensive contact with the nearby Amdo Tibetan (Zhou 2019a, 2019b, 2020a). A number of other syntactic behaviors in Zhoutun, such as the "N+Num" structure (Zhou 2020b), the "locutor-referential pronoun" (Zhou 2021), and the copular system (Zhou 2022a), are also suggested to be induced by the contact with Amdo Tibetan. For a systematic description of Zhoutun, one can refer to Zhou (2022b). In this paper, we demonstrate once more the deep influence of Amdo Tibetan on Zhoutun by illustrating the situation of comparative constructions in

Zhoutun. Moreover, we expect that the case study of Zhoutun (together with Chinese and Amdo Tibetan) will offer a theoretical contribution to the typological study of comparative constructions.

In addition to Zhoutun, there are a number of Chinese dialects (referred to as "Gan-Qing dialects" henceforth) that have been influenced by the surrounding Amdo and Altaic languages (Mongolic: Santa, Bonan, and Mongghul, and Turkic: Western Yugur and Salar) in northwest China, particularly in the west of Gansu, the east of Qinghai, and the border between the two provinces. This area is known as Gan-Qing linguistic area (also referred to as "Qinghai-Gansu Sprachbund" and "Amdo Sprachbund" in the literature, e.g., Slater 2003 and Sandman 2016, respectively). Numerous scholars (Dwyer 1995; Slater 2003; Sandman 2016; Xu 2017; Peyraube 2018; Xu and Peyraube 2018; Zhou 2019a; among others) argue that language contact plays a role in the explanation of a given phenomenon in languages or dialects in this area.

The paper is structured as follows: Section 2 describes five comparative constructions in Zhoutun. Section 3 demonstrates that the comparative constructions in Zhoutun, on the one hand, have features from both Amdo Tibetan and Chinese, and, on the other hand, have a feature that is distinct from both Amdo Tibetan and Chinese, which is the result of contact between the two languages. Section 4 makes a conclusion. We suggest that the typological study of comparative constructions should take into account two parameters: the distinction between comparative subject and attributive subject, and the measurement of the result, which can at least be divided into abstract measurement and concrete measurement.

The data pertaining to Zhoutun presented in this study were acquired through a series of my fieldwork excursions conducted over the course of approximately five months, between September 2014 and October 2020, in the village of Zhoutun. Two distinct sources of data were utilized: structured interviews and naturally occurring discourse. The latter comprised a diverse range of communicative genres, including topic-specific interviews, storytelling, and casual conversations.

## 2. Comparative Constructions in Zhoutun

Prior to further discussion, I will introduce two pairs of terms used in this paper. The first pair of terms are "comparative subject (ComS)" and "attributive subject (AttS)". These two subjects are further subcategorizations of the CS mentioned previously. The ComS corresponds syntactically to the St, while the AttS is the semantic subject of the adjective in the CR. The second pair of terms are "abstract measurement (AbsM)" and "concrete measurement (ConM)". These two terms are further subcategorizations of the CR, where AbsM is a concrete numerical measurement, and ConM is an abstract degree of the measurement. This will be illustrated with examples from Mandarin Chinese:

(3)　a.　头发我比你长一些。

| toufa | wo | bi | ni | chang | yixie. |
|-------|-----|-----|-----|-------|--------|
| hair | 1 | CM | 2 | long | a.little |

'My hair is a little longer than yours.'

　　b.　价格这家商店比那家贵五块。

| jiage | zhe | jia | shangdian | bi | na | jia gui | wu kuai. |
|-------|------|-----|-----------|-----|---------|---------|----------|
| price | this | CL | shop | CM | that CL | expensive | five yuan |

'The price at this shop is five yuan more expensive than that one.'

In (3a), toufa '*hair*' is identified as the AttS as it serves as the semantic subject of the CR expressed by *chang* 'long' (one can say *toufa chang* 'the hair is long'). *Wo* 'I' is classified as the ComS due to its syntactic equivalence with the St *ni* 'you', forming a matching relationship. Analogously, in (3b), *jiage* 'price' is designated as the AttS and *zhe jia shangdian* "this shop" as the ComS. *Yixie* 'a litttle' in (3a) is categorized as the AbsM, while *wu kuai* 'five yuan' in (3b) is considered as the ConM.

### 2.1. xa-Cxn

*xa*-Cxn stands for the comparative construction employing the dative–accusative *xa* (and its allophone *a*) as the CM. As a multifunctional case marker, *xa* serves as a means of marking not only the St but also a variety of semantic roles, including patient, recipient, beneficiary, addressee, possessor, and experiencer. This is demonstrated in (24) of Section 3. *Xa* may have originated from the Chinese locative *xia* 'down' (Zhou 2019b).

This section specifically focuses on *xa*'s function as a CM. Illustrative examples are provided in (3) through (5).

(3) 个箇啊大着个
| kɤ | kua | ta | tʂɤ | kɤ. |
|----|-----|-----|-----|-----|
| this | that:CM | big | PROG | PART |

'This is bigger than that.'

(4) 安文栋连珺哈三岁大着哩。
| āuɤʈū | liātɕỹ | xa | sā | suɨ | ta | tʂɤ | li. |
|-------|--------|-----|------|------|-----|-----|-----|
| A | L | CM | three | year | old | PROG | PART |

'Anwendong is three years older than Lianjun.'

(5) 我你啊岁数大着多。
| ŋɤ | nia | suɨfu | ta | tʂɤ | tuɤ. |
|----|-----|-------|-----|-----|------|
| 1 | 2:CM | age | old | COMP | much |

'I am much older than you.'

*xa*-Cxn is the most common and grammaticalized comparative construction in Zhoutun. Its St–CR word order is in harmony with that of an OV language (Dryer 1992).

In typological studies on comparatives, CS and CR are two essential parameters (Dixon 2012; Stassen 2013; Stolz 2013). *xa*-Cxn presents two features on these two parameters that merit additional discussion. First, *xa*-Cxn involves two categories of comparative subject, which are referred to, following Liu (2012), as "comparative subject" (ComS) and "attributive subject" (AttS) in this paper. The two sorts of subjects can be identical, as seen by *kɤ* 'this' in (3): *kɤ* and *kuɤ*[1], the St, are both demonstrative pronouns, i.e., they have the same syntactic status, and *kɤ* is the subject of the adjective *ta* 'big' (one can say *kɤta* 'this is big'). However, in *xa*-Cxn, the ComS and AttS can be distinct. In (5), for example, the ComS is *ŋɤ* 'I' (syntactically corresponds to the St *ni* 'you'), while the AttS is *suɨfu* 'age', given that it is *suɨfu* rather than *ŋɤ* that serves as the semantic subject of the adjective *ta* in *suɨfu ta* 'the age is old'[2]. In a number of instances, the ComS could be omitted, resulting in the AttS and the St being the objects being compared. See (6).

(6) 你的鞋我啊一号大着哩。
| ni | tɤ | xɛ | ŋa | i | xɔ | ta | tʂɤ | li. |
|----|-----|-----|-----|-----|-----|-----|-----|-----|
| 2 | GEN | shoe | 1:CM | one | size | big | PROG | PART |

'Your shoes are one size bigger than mine.'

(6) has no ComS. If there were a ComS, it should be *ni* 'you' that syntactically corresponds to the St *ŋɤ* 'I'. The AttS is *ni tɤxɛ* 'your shoes' (as in *ni tɤxɛ ta* 'your shoes are big'). This "AttS-St" is ungrammatical in many languages. In English, for example, it would be ungrammatical to convert the English translation in (6) to the "AttS-St" frame (*your shoes are one size bigger than me*, in which AttS= *your shoes* and St= *me*).

In typological studies, the distinction between ComS and AttS has received little attention. This distinction is not addressed in works examining comparative constructions from a typological perspective, such as those by Dixon (2012), Stassen (2013), and Stolz (2013). However, this phenomenon is pervasive in Chinese and Tibetan; see Section 3.

Another feature of *xa*-Cxn is that the measurement of the CR can be further subdivided into the "abstract measurement" (AbsM) and the "concrete measurement" (ConM). The semantic distinction between the two types of measurement is inherent, while Zhoutun distinguishes them linguistically: the AbsM follows the adjective of the CR, whereas the ConM precedes it. For example, in (4) the ConM *sā suɨ* 'three years' precedes the adjective *ta* 'old', while in (5) the AbsM *tuɤ* 'much' occurs after *ta* as a complement. Note that

the English translations reveal that the two types of measurement are all placed before the adjective in English, as in *three years/ much older*. In Mandarin Chinese, both types of the measurement come after the adjective. See (7).

(7) a. 我比你大三岁。

| wo | bi | ni | da | san | sui. |
|----|----|----|----|-----|------|
| 1 | CM | 2 | old | three | year |

   'I am three years older than you.'

  b. 我比你大得多。

| wo | bi | ni | da | de | duo. |
|----|----|----|----|-----|------|
| 1 | CM | 2 | old | COMP | much |

   'I am much older than you.'

In Section 3, I will argue that the different position of ConM and AbsM in Zhoutun reflects the hybrid feature of Chinese and Tibetan and represents an "incomplete" transition from the former to the latter. A preliminary observation shows that languages prefer to place the two types of measurement in the same position relative to the adjective. This is a potential explanation for why previous typological studies did not distinguish between ConM and AbsM. In this light, the data in Zhoutun are worthy of consideration.

*2.2. pi-Cxn*

*pi*-Cxn refers to the comparative construction where *pi* is the CM. *pi* is clearly a Chinese word meaning 'to compare' and serves as the comparative marker in Mandarin Chinese. See some examples of *pi*-Cxn in Zhoutun.

(8) a. 老王比老张头一个高着个。

| lɔuɑ̃ | pi | lɔtʂɑ̃ | thɯɯ | I | kɤ | kɔ | tʂɤ | kɤ. |
|-------|-----|-------|------|----|----|----|----|-----|
| Old.W | CM | old.Z | head | one | CL | tall | PROG | PART |

   'Old Wang is one head taller than Old Zhang.'

  b. 我比你大一岁。

| ŋɤ | pi | ni | ta | i | suɨ. |
|----|----|----|----|----|------|
| 1 | CM | 2 | old | one | year |

   'I am one year older than you.'

(9)  我比你大着多。

| ŋɤ | pi | ni | ta | tʂɤ | tuɤ. |
|----|----|----|----|-----|------|
| 1 | CM | 2 | old | COMP | much |

   'I am much older than you.'

(10)  衣裳的样子你的比我的好看着个，价格我的比你的便宜着个。

| iʂɑ̃ | tɤ | iɑ̃tsi | ni | tɤ | pi | ŋɤ | tɤ | xɔkhã | tʂɤ | kɤ, |
|------|----|--------|----|----|----|----|----|-------|-----|-----|
| cloth | GEN | style | 2 | GEN | CM | 1 | GEN | nice-looking | PROG | PART |

| tɕiakɤ | ŋɤ | tɤ | pi | ni | tɤ | phiãi | tʂɤ | kɤ. |
|--------|----|----|----|----|----|-------|-----|-----|
| price | 1 | GEN | CM | 1 | GEN | cheap | PROG | PART |

   'The style of your clothes is better than mine, and the price of mine is cheaper than yours.'

Some essential properties of (8b), (9), and (10) are likewise present in *pi* comparatives of Mandarin Chinese. See the parallel expression in Mandarin Chinese.

(11)  a.    我比你大一岁。

| wo | bi | ni | da | yi | sui. |
|---|---|---|---|---|---|
| 1 | CM | 2 | old | one | year |

'I am one year old than you.'

    b.    我比你大得多。

| wo | bi | ni | da | de | duo. |
|---|---|---|---|---|---|
| 1 | CM | 1 | old | COMP | much |

'I am much older than you.'

    c.    衣服的样子你的比我的好看，价格我的比你的便宜。

| yifu | de | yangzi | ni | de | bi | wo | de | haokan, |
|---|---|---|---|---|---|---|---|---|
| cloth | GEN | style | 2 | GEN | CM | 1 | GEN | nice-looking |
| jiage | wo | de | bi | ni | de | pianyi | | |
| price | 1 | GEN | CM | 2 | GEN | cheap | | |

The style of your clothes is better than mine, and the price of mine is cheaper than yours.'

On the one hand, both of the ConM *i suɨ* 'one year' in (8b) and the AbsM *tuɤ* 'much' in (9) follow the adjective. On the other hand, the AttS can be distinguished from ComS, as shown in (10) where the AttS is *iʂɑ̃ tɤiɑ̃tsi* 'cloth's style' and *tɕiakɤ* 'price', and the ComS is *ni tɤ* 'yours' and *ŋɤtɤ* 'mine' in the first and second clause, respectively. (11) demonstrates that the parallel *pi* constructions are also seen in Mandarin Chinese. Contrary to Mandarin Chinese *pi* comparatives, however, the ConM in Zhoutun's *pi*-Cxn can also be placed before the adjective, as shown in (8a). This performance can be viewed as the effect of contact with Amdo Tibetan, where the ConM comes before the adjective; see Section 3.

In general, *pi*-Cxn is employed less frequently than *xa*-Cxn to convey comparative meaning. However, the author's empirical observation suggests that the frequency of usage is on the rise. A comparison of the author's fieldwork in 2014 and 2020 indicates that the utilization of *pi*-Cxn has become more prevalent and intuitive[3]. This is likely due to the increasing influence from Mandarin Chinese nowadays. It is unknown whether *pi*-Cxn will eventually exceed and supplant *xa*-Cxn. Nonetheless, it is evident that the language contact is not static and is ongoing.

*2.3. 'look'-Cxn*

'look'-Cxn uses the verb khã 'look' to introduce the St, forming the structure "St-*kh*ã, CS-CR". For example,

(12)  你们的房子看时，我们的房子大。

| ni | mɤ | tɤ | fɑ̃tsi | khã | ʂi, | ŋɤ | mɤ | tɤ | fɑ̃tsi | ta. |
|---|---|---|---|---|---|---|---|---|---|---|
| 2 | PL | GEN | house | look | if | 1 | PL | GEN | house | big |

'My house is bigger than yours (lit. looking at your house, my house is bigger).'

(13)  安文栋的分数看了嘀，连珺的分数高哩。

| ãuɤtõ | fɤ�Ɂu | kha | lɔ | ti, | liãtɕỹ | tɤ | fɤɁu | kɔ | li. |
|---|---|---|---|---|---|---|---|---|---|
| A | GEN | scorelookPFV | | PARTL | | GEN | score high | | PART |

'Lianjun's score is higher than Anwendong's. (lit. looking at Anwendong's score, Lianjun's is higher).'

The literal meaning of 'look'-Cxn is "(If) looking at X, Y is Adj", and it implicates the comparative meaning "Y is Adj than X". The conditional *ʂi* (time>when>if) is optional. Moreover, no instance is found within which the CS splits into ComS and AttS: they are identical in 'look'-Cxn (at least in my data). In the data I also find no example dealing with the measurement of the CR. Instead, bare adjective is used in this construction.

*2.4. 'and'-Cxn*

'and'-Cxn refers to the structure "X-and-Y(-verb), X/Y-CR", conveying the comparative meaning "X/Y is adjective than Y/X". This construction is similar to 'look'-Cxn in that both of them consist of two clauses, and the first clause provides the object(s), and the second clause shows the comparative result. For example:

(14) 羊肉带大肉，羊肉香着个。

| iɑ̃ȵ̃u | tɛ | taȵ̃u, | iɑ̃ȵ̃u | ɕiɑ̃ | tʂɤ | kɤ. |
|---|---|---|---|---|---|---|
| mutton | CONJ | pork | mutton | fragrant | PROG | PART |

'Mutton is more delicious than pork (lit. between mutton and pork, mutton is delicious).'

(15) 我带扎西两个，扎西俊哩喔。

| ŋɤ | tɛ | tʂaɕi | liɑ̃ | kɤ, | tʂaɕi | tɕỹ | li | uɤ. |
|---|---|---|---|---|---|---|---|---|
| 1 | CONJ | Z | two | CL | Z | good.looking | PARTPART |  |

Zhaxi is more handsome than me (lit. between Zhaxi and I, Zhaxi is handsome).'

(16) 我馍馍带面片儿吃时，馍馍吃着多。

| ŋɤ | mɤmɤ | tɛ | miɑ̃phiɛ | tʂhi | ʂi | mɤmɤ | tʂhi | tʂɤ | tuɤ. |
|---|---|---|---|---|---|---|---|---|---|
| 1 | steamed.bread | CONJ | noodle | eat | if | steamed.bread | eat | COMP | more |

I eat more steamed bread than noodles (lit. between eating steamed bread and noodles, I eat more steamed bread).'

Although comparable to 'look'-Cxn in some respects, 'and'-Cxn differs significantly from 'look'-Cxn and the other two comparative constructions, namely, *xa*-Cxn and *pi*-Cxn. That is, 'and'-Cxn is a loosely structured comparative strategy rather than a dedicated comparative construction. First, in the structure "X-and-Y(-verb), X/Y-CR", the subject of the CR cannot be determined until the second clause appears. Changing *iɑ̃ȵ̃u* 'mutton' in the second clause of (14) to *taȵ̃u* 'pork' would result in the sentence 'Pork is more delicious than mutton', in which 'pork' becomes the subject, as opposed to the original version, where 'mutton' is the subject. Second, since 'and' is not a comparative marker, the coordinating construction in the first clause does not necessitate further comparison in the second clause. In (17), for instance, the coordination *ŋɤtɛ tʂaɕi* 'I and Zhaxi' functions as the agent of the verb *tɕhi* 'go' in the second clause, and no comparative sense is indicated.

(17) 我带扎西两个，我两个学里去了。

| ŋɤ | tɛ | tʂaɕi | liɑ̃ | kɤ, | ŋɤ | liɑ̃ | kɤ | ɕyɤ | li | tɕhi lɔ. |
|---|---|---|---|---|---|---|---|---|---|---|
| 1 | CONJ | Z | two | CL | 1 | two | CL | school | in | go PFV |

'Zhaxi and I went to school.'

Nevertheless, 'and'-Cxn is taken into account while discussing comparative expressions for two reasons. First, this construction is a common and natural manner of conveying comparative meaning in Zhoutun. Moreover, there is a possibility that 'and'-Cxn will evolve into a dedicated comparative construction (such a process can be found in Wu dialects, as discussed in Section 3). Second, 'and'-Cxn can be used to compare VPs, which is not possible with any of the aforementioned constructions. This is evidenced by (16). The two objects compared in (16) are the NPs within the VP, i.e., *mɤmɤ* 'steamed bread' and *miɑ̃phiɛ* 'noodle'. This function, i.e., comparing NPs within the VP, cannot be satisfied by *xa*-Cxn, *pi*-Cxn, and 'look'-Cxn, and is actually prohibited in Mandarin Chinese *pi* comparatives. Let us examine *pi* comparatives in Mandarin in (18)–(19).

(18)    a.

| Zhangsan | rou | chi | de | bi | fan | duo. |
|---|---|---|---|---|---|---|
| Z | meat | eat | COMP | CM | rice | more |

     b.   张三肉比饭吃得多。

| Zhangsan | rou | bi | fan | chi | chi | duo. |
|---|---|---|---|---|---|---|
| Z | meat | CM | rice | eat | COMP | more |

'Zhangsan eats meat more than rice.'

(19)    a.   * 张三比肉更爱吃米饭。

| Zhangsan | bi | rou | geng | ai | chi | mifan. |
|---|---|---|---|---|---|---|
| Z | CM | meat | more | like | eat | rice |

     b.   张三肉比米饭更爱吃。

| Zhangsan | rou | bi | mifan | geng | ai | chi. |
|---|---|---|---|---|---|---|
| Z | meat | CM | rice | more | like | eat |

'Zhangsan likes to eat meat more than rice.'

(18) demonstrates that the CS and St can be the patient NPs of the verb, such as the *rou* 'meat' and *fan* 'rice' in (18a) and (18b). Note that these NPs should precede the adjective

CR. Another sort of comparative construction in which the CR is a verb is seen in (19). In this construction, neither CS nor St are placed after the verb CR, the typical position of an object in Mandarin.

Liu (2012) contends that Mandarin *pi* comparative NP$_{CS}$-NP$_{St}$-Adj/V$_{CR}$ observes a syntactic restriction, namely the "topic-domain restriction": in [NP$_{CS}$-NP$_{St}$]$_{topic}$-[Adj/V$_{CR}$]$_{predicate}$, the CS and St must be constrained in the topic domain of the clause, i.e., before the CR. Consequently, no elements in the predicate domain may be directly compared. Thus, it is ungrammatical to directly compare the NP after the verb, as this would violate the topic-domain restriction if the [NP$_{CS}$]$_{topic}$-[V-NP$_{St}$]$_{predicate}$ were used.

In fact, the topic-domain restriction is observed not only at the clause level, but also in the VP that is before the adjective CR. As evidenced by (20), the comparative expression [V-NP$_{CS}$-NP$_{St}$]-[Adj] is ungrammatical.

(20)  * 张三吃肉比饭多。

| Zhangsan | chi | rou | bi | fan | duo. |
|---|---|---|---|---|---|
| Z | eat | meat | CM | rice | more |

'Zhangsan eats meat more than rice.'

Only when *rou* 'meat' or both *rou* and *fan* 'rice' are placed before the verb, as shown by (19), would (20) become grammatical. Even if [V-[NP$_{CS}$-NP$_{St}$]] may be viewed as the topic domain relative to [Adj] as the predicate, [V-[NP$_{CS}$-NP$_{St}$]] itself requires some sort of topic-domain limitation in that at least the NP$_{CS}$ must be placed before the verb, the topic position relative to the verb.

The topic-domain restriction is not exclusive to *pi* comparative. Mandarin could use structures akin to the 'and'-Cxn in Zhoutun to express comparative meaning. The coordinated NPs could not be placed after the verb in this 'and'-like comparative; see (21a).

(21)  a.  * 我吃馍馍和面条，馍馍吃得多。

| wo | chi | momo | he | miantiao | momo | chi de | duo. |
|---|---|---|---|---|---|---|---|
| 1 | eat | steamed.bread | CONJ | noodles | steamed.bread | eat COMP | more |

   b.  我馍馍和面条，馍馍吃得多。

| wo | momo | he | miantiao | momo | chi de | duo. |
|---|---|---|---|---|---|---|
| 1 | steamed.bread | CONJ | noodles | steamed.bread | eat COMP | more |

'I eat more streamed bread than noodles (lit. between steamed bread and noodles, I eat more steamed).

The "steamed bread" and "noodles" in (22a) are located in the predicate domain (after the verb), making the clause ungrammatical. By simply omitting the verb, (21b) becomes grammatical, as the two NPs are identified as the topic.

In contrast to Mandarin, Zhoutun 'and'-Cxn permits a direct comparison between two NPs within the VP. Comparing (16) to (22a), this discrepancy is readily apparent. It does not mean, however, that the topic-domain restriction fails in Zhoutun. Since Zhoutun's basic word order had shifted to SOV, the O occupies both the topic domain in terms of the clause and the predicate domain in terms of the verb: [S-O]$_{topic}$-[V]$_{predicate}$ or [S]$_{topic}$-[[O-V]$_{VP}$]$_{predicate}$. The dual identity of O enables the acceptability of the structure [S-O$_{NP1-and-NP2}$-V], in which NP$_1$-and-NP$_2$ observes the topic-domain restriction as a topic, and occur within the VP.

### 2.5. Hybrid Cxn

Some of the mentioned constructions can be used simultaneously, forming what we term the "hybrid construction". There are two common hybrid constructions. First, the hybrid of 'and' and *xa*. For example:

(22)  我带扎西两个，扎西我啊大着个。

| ŋɤ | tɛ | tʂaɕi | liã̃ | kɤ, | ŋa | ta | tʂɤ | kɤ. |
|---|---|---|---|---|---|---|---|---|
| 1 | CONJ | Z | two | CL | 1:CM | old | PROG | PART |

'Between Zhaxi and I, Zhaxi is older than me.'

Another type of hybrid construction consists of *xa*, 'look' and *pi*. For example,

(23)　　　我语文哈看了着，比数学好。

| ŋɤ | yuʏ̃ | xa | khã | lɔ | tʂɤ, | pi | fuɕyʏ | xɔ. |
|----|------|----|-----|----|------|----|-------|-----|
| 1 | Chinese | CM | look | PFV | PROG | CM | math | good |

'Looking at my Chinese, it is better than my math.'

In sum, the comparative constructions in Zhoutun can be concluded as in Table 1.

**Table 1.** The types and forms of the comparative constructions in Zhoutun.

| Type | Form |
|------|------|
| *xa*-Cxn | CS(AttS-ComS)+St-*xa*+(ConM+)CR<br>CS(AttS-ComS)+St-*xa*+CR(+AbsM) |
| *pi*-Cxn | CS(AttS-ComS)+*pi*-St+CR(ConM/AbsM)<br>CS(AttS-ComS)+*pi*-St+(ConM+)CR |
| 'look'-Cxn | St-'look', CS+CR |
| 'and'-Cxn | X-'and'-Y(-verb), X/Y-CR[4] |
| Hybrid Cxn | 'and'-Cxn plus *xa*-Cxn<br>*xa*-Cxn plus 'look'-Cxn plus *pi*-Cxn |

## 3. Hybrid Features in Comparative Constructions of Zhoutun

Comparative constructions in Zhoutun exhibit both Tibetan and Chinese characteristics. The hybrid features provide an intriguing illustration of how Zhoutun's grammar has been shaped by language contact. We will discuss each one individually.

First, *xa*-Cxn. Zhoutun's SOV order and case marking system are undoubtedly induced by contact with Amdo Tibetan. As a dative–accusative marker, *xa*, can be employed to indicate a variety of semantic roles; see (24).

(24)　a.　我箇啊书一本给了。

| ŋɤ | kua | fu | i | pɤ̃ | kɨ | lɔ. |
|----|-----|----|----|-----|----|-----|
| 1 | 3:DAT | book | one | CL | give | PFV |

'I gave him/her a book'.

　　b.　扎西玉林哈衣裳取给。

| tʂaɕi | ylĩ | xa | iʂɑ̃ | tshɯ | kɨ. |
|-------|-----|----|------|------|-----|
| Z | Y | DAT | coat | take | give |

'Zhaxi takes the coat for Yuli'.

　　c.　扎西我啊说着个

| tʂaɕi | xa | tɕhiɛ | iɯti |
|-------|----|-------|------|
| Z | DAT | money | have PART |

Zhaxi has money'.

　　d.　扎西哈钱有嘀。

| tʂaɕi | xa | ɻɤ | tʂɤ | xɤ̃ | lɨ. |
|-------|----|----|-----|-----|-----|
| Z | dat | hot | comp | very | part |

'Zhaxi feels very hot'.

　　e.　扎西哈热着很哩。

| tʂaɕi | xa | ɻɤ | tʂɤ | xɤ̃ | lɨ. |
|-------|----|----|-----|-----|-----|
| Z | | dat | hot | comp | very part |

Zhaxi feels very hot'.

　　f.　扎西玉林哈打了。

| tʂaɕi | ylĩ | xa | ta | lɔ. |
|-------|-----|----|----|-----|
| Z | Y | ACC | beat | PFV |

'Zhaxi has beaten Yulin'.

(24) illustrates the semantic roles that can be marked by *xa/a*, including the recipient (24a), benefactor (24b), addressee (24c), possessor (24d), experiencer (24e), and patient (24f).

Since Altaic languages are nominal–accusative and Tibetan is ergative–absolutive, some believe that *xa* in Gan-Qing dialects was formed due to the influence from Altaic languages (e.g., Zhang 2013). However, this theory fails to account for the following two crucial performances: (1) while Altaic languages lack dative–accusative syncretism[5], all Gan-Qing dialects utilize *xa* as a dative–accusative marker; and (2) only Tibetan uses dative to mark the experiencer (i.e., the subject who experiences particular feelings such as 'hot' and 'angry'). I argue that contact with Tibetan is the cause of the emergence of *xa*, the case marking system, and the SOV order in Gan-Qing dialects[6]; for details, see Zhou (2019a, 2019b, 2020a).

*xa*-Cxn conforms to the typical word order of comparative constructions in OV languages, namely St-CR (Dryer 1992). This OV-feature of *xa*-Cxn is a result of contact with Tibetan. In Amdo Tibetan, the word order of a comparative construction is identical to that of *xa*-Cxn (without considering the abstract measurement; see below), as shown in (25).

(25)　　tsheɾuɯŋ　　　wsonam-ma-wtina　　　lo　　　　tɕhe-gɯɪ.
　　　　　C　　　　　　S-dat-cm　　　　　　　age　　　old-aux
　　　　　Cairang is elder than Sunnanmu. (Shao 2012, p. 29)

In addition to the general word order of *xa*-Cxn, another feature of the Tibetan type is that the CR's concrete measurement precedes the adjective. See (4) in Zhoutun (repeated in (26a)) and (26b) below in Amdo Tibetan.

(26)　a. 安文栋连珺哈三岁大着哩。
　　　　　ãuɣtũ　　liãtɕỹ　　xa　　　sã　　sui̵　　ta　　tʂɣ　　li.
　　　　　A　　　　L　　　　cm　　three　year　old　prog　part
　　　　　'Anwendong is three years older than Lianjun.'
　　　b. ŋa　　　ɸtʂaɕʰi=ʔa　ɸti=na　　mi　　　khərzək=kə　rəŋ=ŋɡə.
　　　　　1　　　Z=all　　　look=cond　meter　one=erg　　tall=mir
　　　　　'I am one meter taller than Zhaxi.' (Mingyuan Shao, p.c.)

In contrast, the concrete measurement of the CR should follow the adjective in a Chinese comparative construction; see (7a) and (27).

(27)　　我比扎西高一米。
　　　　　wo　　　bi　　　Zhaxi　　gao　　yi　　　mi.
　　　　　1　　　cm　　　Z　　　　tall　　one　　meter
　　　　　'I am one meter taller than Zhaxi.'

However, when it comes to the abstract measurement of the CR, the scenario changes: the abstract measurement in Zhoutun, which is placed after the adjective, maintains the same position as in Chinese (28a), but differs from Tibetan (28b).

(28)　a.　我比扎西高得多。
　　　　　wo　　　bi　　　Zhaxi　　gao　　de　　　duo.
　　　　　1　　　cm　　　Z　　　　tall　　comp　much
　　　b.　ŋa　　　ɸtʂaɕhi=ʔa　ɸti=na　　jərzək=kə　rəŋɡə.
　　　　　1　　　Z=all　　　look=cond　much=erg　　rəŋɡə.
　　　　　'I am much taller than Zhaxi.' (Shao, p.c.)

As demonstrated by the aforementioned examples, with regards to the position of AbsM and ConM, the differences between comparative constructions in Chinese and Tibetan are reflected in the type of word order, with Chinese exhibiting a mirror image of Tibetan in this aspect. Specifically:

Chinese: CS-St-CR-AbsM/ConM
Tibetan: CS-St-AbsM/ConM-CR.

Interestingly, the performance of *xa*-Cxn in Zhoutun in this regard is not entirely comparable to either Chinese or Tibetan. The word order of *xa*-Cxn generally reflects the Tibetan type in that it has St-CR order, with the ConM preceding the CR. In terms of the position of AbsM, however, *xa*-Cxn retains a Chinese feature in that the abstract measurement follows the adjective as a complement. That is:

Zhoutun: (1) CS-St-CR-AbsM; (2) CS-St-ConM-CR

Second, *pi*-Cxn. It is evident that *pi*-Cxn corresponds to its Chinese counterpart, which we refer to as "*pi* comparatives". First, both *pi*-Cxn and *pi* comparatives put abstract and concrete measurement after the adjective of the CR. Second, they both distinguish ComS and AttS. The only difference between the two is that in *pi*-Cxn, the concrete measurement could also be put before the adjective, as shown by (8a). This feature is distinct from the *pi* comparatives and has to do with Amdo Tibetan, in which concrete measurement occurs before the CR. The word order between St and CR in *pi*-Cxn is the same as in Chinese, namely St-CR. Interestingly, this "Chinese type" word order itself is in accordance with OV. In fact, the order of St-CR in Chinese *pi* comparatives is abnormal compared to the typical CR-St in VO languages (Dryer 1992). Given that Zhoutun had changed its basic word order into SOV, the original "abnormal" word order of *pi* comparatives in Chinese[7] became "normal" in Zhoutun *pi*-Cxn.

The contrast between the comparative subject (ComS) and the attributive subject (AttS) is a significant feature shared by both the *xa*-Cxn and the *pi*-Cxn. The distinction between these two types of subject, according to Liu (2012), shows the topic prominence of Chinese comparative constructions. As a topic-prominent language (Li and Thompson 1981), the topic in a Chinese sentence may not have an "argument relationship" with the predicate (i.e., the topic is not an argument of the predicate), so long as some relevance exists, even if it is blurred. See the renowned example in (29).

(29)    那场火幸亏消防队来得快

nei chang huo xingkui        xiaofangdui lai        de            kuai.
that CL fire fortunate        fire.brigade come      COMP         fast
'That fire, fortunately the fire brigade came quickly.'

In this sentence, *nei chang huo* 'that fire' is not an argument of the predicate *lai de kuai* 'come quickly'. This trait is also present in Chinese *pi* comparative, in which the ComS and CR can have no argument relationship. See (30)

(30)   a.   他房子比我贵。
            ta            fangzi        bi           wo           gui.
            3             house         CM           1            expensive
       b.   房子他比我贵。
            fangzi        ta            bi           wo           gui.
            house         3             CM           1            expensive
            'His house is more expensive than mine (/ *me/*I).'
            (*He house is more expensive than I/ * House, he is more expensive than I.)

Regardless of the position, *fangzi* 'house' and *ta* 'he' are separated in (30). As the argument of the adjective *gui* 'expensive', *fangzi* is the AttS, whereas *ta* is the ComS that is syntactically compared to the St *wo* 'I'. ComS and St cannot serve as the subject of the adjective, as in *ta/wo gui* 'he/I expensive'. Note that in English, the ComS should also be the AttS, that is, both roles should be represented by one and the same NP. The English translation in (30) illustrates this performance clearly: *his house* is the ComS in the sense that it is the argument of the adjective *expensive*, and it is the AttS in the sense that it is syntactically compared to the St *mine*.

Following Liu, Shao (2012) observed that the ComS and AttS are also splitable in the comparative constructions in Amdo Tibetan. For example,

(31) ɸigoɾmo　cçʰo-a-wtina　ŋa　ŋoŋ-nguɪ, xʰiwɕa　cçʰo-a-wtina　ŋa　maŋ-nguɪ.
money　2-DAT-CM　1　less-AUX　knowledge　2-DAT-CM　1　more-AUX
'My money is less than yours; my knowledge is more than yours.'
(lit. As for money, I have less than you; as for knowledge, I have more than you.)

In earlier research, I had believed that the distinction between the ComS and the AttS in comparative constructions of Zhoutun was a Chinese reservation. However, Shao's research reveals that Amdo Tibetan has this characteristic. Therefore, it cannot be ruled out that Tibetan has impacted it. Regardless of its origin, the distinction between the two types of subject in Sino-Tibetan languages is an important topic deserving of more study. Researchers of Sino-Tibetan languages can investigate if other Sino-Tibetan languages share this characteristic; typologists can include ComS and AttS in cross-linguistic studies of comparative constructions.

Now we turn to 'look'-Cxn. 'look'-Cxn is found not just in Zhoutun, but also in other Gan-Qing dialects, as well as Amdo Tibetan and Altaic languages in the Gan-Qing linguistic area, as discussed in Sandman and Simon (2016). For example:

(32) a. Amdo Tibetan
lhasa-'a　ɸti-na　sələŋ　tʃhe-gi.
Lhasa-DAT　look-COND　Xining　big-TEST
'Xining is bigger than Lhasa.'
b. Wutun
je-ge　jjhakai　zhungo　kan-la　xaige　ga-li.
this-REF　country　China　look-COND　very　small-SEN.INF
'This country is much smaller than China.'
c. Salar
biqirox　jiguo elige　ʥan-aŋ　vaq-sə　da　aɣər-a　ro.
cloth　all　that.way life-2POSS look-COND too　heavy-TEST　INT
Are all such clothes weightier (i.e., more important) than your life?'

They assert that, at first glance, the 'look' comparatives are derived from Amdo Tibetan, as Tibetan is what they refer to as the "model language" and has a significant impact on both Altaic languages and Gan-Qing dialects in the same area in a number of respects. However, as 'look' comparatives are not found in any Tibetan language outside of the Gan-Qing linguistic area, Sandman and Simon believe that 'look' comparatives are an independent development of the Gan-Qing linguistic area and their source "remains unclear" (p. 112). The development of comparative markers from "look" in language is well documented, as one of the reviewers pointed out, Korean being a case in point. According to Rhee (2022), the grammaticalization source of the Korean comparative marker "pota" is "look".

Although the synchronic evidence in Gan-Qing linguistic area may not lead to a definitive conclusion regarding the origin of 'look' comparatives, diachronic data offer a clue. Shao (2012) illustrates the presence of 'look' comparatives in classic Tibetan, as in (33).

(33) da　ro-las-bltasna　ni　ŋamtɕhuŋ,
DEM　corpse-CM-look　TOP　weak
rma-las-bltasna　phogsŋa　tɕhe-ʑing-mtɕhiis-na.
wound-CM-look　scar　big-AUX-AUX-have
'(I am) weaker than a corpse; the scar is bigger than the wound.' (*Dunhuang Historical Documents of Tubo*, P.T. 1287 P1.566; Shao 2012, p. 30)

Shao (2012) proposes that the widespread use of 'look' comparatives in contemporary Amdo is a remnant of classic Tibetan. I argue, based on his convincing evidence, that the 'look'-Cxn in Zhoutun is formed due to the contact with Amdo. A minor distinction between Zhoutun and Amdo is that in Zhoutun, the COND morpheme, namely ʂ*i*, is optional.

The fourth construction that should be discussed is the 'and'-Cxn. As mentioned, it is better to consider the 'and'-Cxn in Zhoutun as a strategy rather than a grammaticalized construction to convey comparative meaning. The strategy consists of the frame "A and B,

A/B is ADJ", which is pragmatically interpreted as the comparative "A/B is more ADJ than B/A". This strategy represents the topic prominence in Chinese, and the same strategy can be found in Wu dialects that have a higher degree of topic prominence than Mandarin. See the example from Shaoxing Wu (Yimin Sheng, p.c.).

(34)　小张作小王麽，总还是小王高些。

| ɕiɔtsaŋ | tso | ɕiɔɦuɔŋ | meʔ, | tsoŋ ɦuɛʔzeʔ | ɕiɔɦuɔŋ | kɔ | seʔ. |
|---|---|---|---|---|---|---|---|
| Little.Z | and | Little.W | TOP | mustCOP | Little.W | tall | a.little |

'As for Little Zhang and Little Wang, Little Wang is a little taller.'

(34) in Shaoxing Wu, like the 'and'-Cxn in Zhoutun, uses the coordination "A and B" as a comparison scope for two objects with the potential to be compared, and the following clause "A/B is ADJ" completes the comparison between A and B. With the usage of the topic marker, "A and B"-"A/B is ADJ" in Shaoxing Wu constitutes an obvious topic–comment pair. Although there is no topic marker in 'and'-Cxn in Zhoutun, the frame "A and B"-"A/B is ADJ" is probably likewise a topic–comment structure. This topic–comment structure in Shaoxing Wu further developed into a grammaticalized construction, as seen in (35a) to (35b).

(35)　a.　小张麽，还是小王高。

| ɕiɔtsaŋ | meʔ, | ɦuɛʔzeʔ | ɕiɔɦuɔŋ | kɔ. |
|---|---|---|---|---|
| Little.Zhang | TOP | COP | Little.Wang | tall |

'Talking about Little Zhang, Little Wang is taller (than him/her).'

　　b.　小张还是小王高。

| ɕiɔtsaŋ | ɦuɛʔzeʔ | ɕiɔɦuɔŋ | kɔ. |
|---|---|---|---|
| Little.Zhang | COP | Little.Wang | tall |

'Little Wang is taller than Little Zhang.'

It is not impossible that the 'and'-Cxn in Zhoutun would undergo the same process to become a dedicated comparative construction in the future, although this has not yet occurred.

The last type of comparative construction that should be discussed is the hybrid Cxn. In this kind of construction, we can see the hybrid features from both Tibetan and Chinese more straightforwardly, as shown by the co-existence between 'and' (Chinese) and *xa* (Tibetan), and between 'look', *xa* (Tibetan) and *pi* (Chinese).

## 4. Conclusions

This paper investigated the comparative constructions in Zhoutun, a Chinese variety that is heavily influenced by Amdo Tibetan. In Zhoutun, there are five comparative constructions: *xa*-Cxn, *pi*-Cxn, 'look'-Cxn, 'and'-Cxn, and hybrid Cxn.

Zhou (2022a) notes that in general Zhoutun possesses hybrid characteristics of Chinese and Tibetan, namely, Chinese phonology and Tibetan syntax. This paper indicated that Zhoutun syntax also expresses the hybrid characteristic. Regarding comparative constructions, this hybrid characteristic might be viewed in the following ways. First, five types of comparative constructions are utilized simultaneously, including *xa*-Cxn and 'look'-Cxn, which are clearly influenced by Amdo Tibetan, and *pi*-Cxn and 'and'-Cxn, from which one can discern the Chinese trace. Second, although the *xa*-Cxn and *pi*-Cxn are generally more Tibetan-like and Chinese-like, respectively, they also contain traces of the other language. For instance, the *xa*-Cxn, which corresponds in word order with Amdo Tibetan, retains a Chinese performance in terms of the position of abstract measurement of the adjective; in contrary, despite the fact that the *pi*-Cxn corresponds to Chinese *pi* comparative in many respects, the Tibetan influence can be discerned from the position of the concrete measurement of the adjective. The distinction between concrete and abstract measurement reflects the transition between Chinese and Tibetan and gives comparative constructions in Zhoutun their own syntactic behavior.

This paper proposed to differentiate between comparative subject and attributive subject, as well as abstract and concrete measurement, thus enhancing our comprehension of

comparative constructions in typological studies. Liu (2012) posits that the distinction between ComS and AttS reflects the topical attributes of the Chinese *pi*-comparatives. As a variant of Chinese, Zhoutun also exhibits this characteristic, which highlights the prominence of its topic structure. Tibetan, as a language with developed topic structure, also displays this feature. According to a reviewer's suggestion, Japanese and Korean, also topic-prominent, seem to display similar features. Hence, the distinction between ComS and AttS could serve as a relevant variable in future cross-linguistic studies on comparative constructions. The distinction between AbsM and ConM, however, has not yet been found in other languages, but the case of Zhoutun suggests that the two could be differentiated in terms of position. This suggests that further attention could be paid to this issue when studying other languages.

**Funding:** The key project of Chinese National Social Science Fund (19AYY004).

**Institutional Review Board Statement:** Not applicable.

**Informed Consent Statement:** Not applicable.

**Data Availability Statement:** Not applicable.

**Acknowledgments:** The author expresses gratitude to the three anonymous reviewers for their valuable comments and insights that have greatly improved this paper. Any remaining issues are solely the responsibility of the author.

**Conflicts of Interest:** The author declares no conflict of interest.

## Notes

[1] *kuɤ+ a→kua*. So do *ni+a→nia*, and *ŋɤ+a→ŋa* below.

[2] *ŋɤta* is also grammatical, unless it does not mean 'I am old' but 'I am big/tall (in figure)'. Also see (1). Thus, in (3), *ŋɤ* is not the semantic subject of *ta*.

[3] In the 2020 investigation, the author observed a more frequent and natural use of *pi*-Cxn by the instructor, compared to the data collected during the 2014 fieldwork. However, this is an empirical observation and further investigation is necessary to establish its validity.

[4] Both X and Y can potentially serve as either the CS or the St, depending on which one forms a combination with the CR in the subsequent clause.

[5] Mongolic languages in the Gan-Qing linguistic area exhibit dative–locative syncretism, whereas Turkic languages exhibit syncretism between dative and allative. In these languages, accusative markers are strictly separate from dative markers. See Zhou (2020a) for details.

[6] It does not necessarily imply that Altaic languages have no effect on Zhoutun and other Gan-Qing dialects. Certain Altaic languages do, in fact, impact Gan-Qing dialects such as Tangwang (Xu 2017) and Gan'gou (Yang and Zhang 2016). However, as argued by Zhou (2020a), the influence should have occurred in the second stratum of language contact in the Gan-Qing linguistic area, whereas the contact between Tibetan and Chinese is earlier and more fundamental for the formation of the current grammatical system of Gan-Qing dialects, particularly for Zhoutun, in which little evidence of the influence of Altaic languages is found.

[7] As noted by one of the reviewers, there are indeed examples of CR-St order in Chinese that are in harmony with the VO order. This is evidenced in the 于 *yu* 'in, at, than' comparatives in ancient Chinese and the 过 *guo* 'surpass' comparatives in Southern dialects such as Cantonese.

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
