# Peer review of "Comparative Constructions in Zhoutun from a Language Contact Perspective"

_languages, doi:10.3390/languages8010066_

Round 1
Reviewer 1 Report
Generally speaking, I find this paper original, clearly structured and well argumented. It presents novel and unique data on previously under-documented Chinese variety spoken in a context of highly multilingual Qinghai-Gansu Sprachbund, so it is a valuable to contribution to Chinese linguistics and contact linguistics. In addition, comparatives in Chinese varieties are also relevant for cross-linguistic typologies of comparatives. I only have some minor suggestions for improvement that mainly concern the contextualization of the study with respect to previous research and presentation of the Zhoutun data.
1. Introduction: Some more information on Zhoutun and its contact with other languages of the Sprachbund (past and present) would be in order in the introduction to help the reader in contextualizing the contact-induced features presented in the paper. For example, it would be good to explicitly mention that Zhoutun is a variety of Northwest Mandarin. A brief summary of the main typological features of Zhoutun would be helpful for the reader, as well as some information of its history of contact with Amdo Tibetan. How about the present language situation? Are speakers of Zhoutun today bilingual in Amdo Tibetan? How about the knowledge of Standard Mandarin today? Is the Zhoutun variety still a stable language or is there any language shift going on? In addition, it would be good to add some explanation on how the data for the paper was collected. Is it based on author's own fieldwork? This becomes somewhat evident later in the paper, but could be described more explicitly in the introduction.
In addition to more explanation on Zhoutun and the data used in the paper, the introduction could include some more references on typologies of comparatives, because the paper aims at contributing the cross-linguistic studies on the topic. Possible typological studies to briefly discuss and cite in the paper include Dixon, R.M.W. 2012. Basic Linguistic Theory. Volume 3. Further Grammatical Topics. Oxford: Oxford University Press. (the chapter on comparatives), as well as Stolz, Thomas 2013. Competing Comparative Constructions in Europe. Berlin: Akademie-Verlag.
2. The numbering of sections: Please check the numbering of sections carefully; in the current paper, section 2 is numbered as section 1.
3. Section 2.1.: Section 2.1. on the second page starts with the examples on comparatives that make use of the dative-accusative marker xa/a. A (very) brief explanation of the functions and origins of this marker would be helpful for the reader, and the Section 2.1 could include a cross-reference to the Section 3 where the various functions of xa/a are explained in more detail.
4. Section 2.2.: Section 2.2. contains a very interesting discussion on comparatives that make use of the element pi, 'to compare'. It would be good to add an example of Standard Mandarin to this section to illustrate more clearly that the form pi in Zhoutun is a cognate to Standard Mandarin, as well as demonstrate the point made in the section that examples (7b), (8) and (9) share some important properties with Standard Mandarin. (This may be obvious for readers who know Mandarin Chinese, but the paper may also be of interest for the readers who are not knowledgeable about Mandarin Chinese grammar).
Section 2.2., second chapter: Mandarin Chinese's pi comparatives > Mandarin Chinese pi comparatives
5. The table 1: The formatting of Table 1 looks somewhat incomplete. Why is the column 'Form' let empty? In addition to type and form of the comparative constructions, it would be helpful for the readers to add a brief summary of origins of each construction.
6. Conclusions: In the conclusions, it would be good to discuss the Zhoutun data more explicitly in relation to typological studies on comparatives. Is the distinction between comparative subject and attributive subject, as well as concrete and abstract measurement lacking from previous typologies altogether?
Author Response
Response to reviewer 1:
Generally speaking, I find this paper original, clearly structured and well argumented. It presents novel and unique data on previously under-documented Chinese variety spoken in a context of highly multilingual Qinghai-Gansu Sprachbund, so it is a valuable to contribution to Chinese linguistics and contact linguistics. In addition, comparatives in Chinese varieties are also relevant for cross-linguistic typologies of comparatives. I only have some minor suggestions for improvement that mainly concern the contextualization of the study with respect to previous research and presentation of the Zhoutun data.
- Introduction: Some more information on Zhoutun and its contact with other languages of the Sprachbund (past and present) would be in order in the introduction to help the reader in contextualizing the contact-induced features presented in the paper. For example, it would be good to explicitly mention that Zhoutun is a variety of Northwest Mandarin. A brief summary of the main typological features of Zhoutun would be helpful for the reader, as well as some information of its history of contact with Amdo Tibetan. How about the present language situation? Are speakers of Zhoutun today bilingual in Amdo Tibetan? How about the knowledge of Standard Mandarin today? Is the Zhoutun variety still a stable language or is there any language shift going on? In addition, it would be good to add some explanation on how the data for the paper was collected. Is it based on author's own fieldwork? This becomes somewhat evident later in the paper, but could be described more explicitly in the introduction.
In addition to more explanation on Zhoutun and the data used in the paper, the introduction could include some more references on typologies of comparatives, because the paper aims at contributing the cross-linguistic studies on the topic. Possible typological studies to briefly discuss and cite in the paper include Dixon, R.M.W. 2012. Basic Linguistic Theory. Volume 3. Further Grammatical Topics. Oxford: Oxford University Press. (the chapter on comparatives), as well as Stolz, Thomas 2013. Competing Comparative Constructions in Europe. Berlin: Akademie-Verlag.
Response: I am grateful for the reviewer's suggestions, which I have incorporated in the revised version. Specifically, in Section 1, I have added an introduction of the background information relevant to Zhoutun. Additionally, I have reviewed the reference materials provided by the reviewer. These materials have been instrumental in my enhanced understanding of the comparative constructions.
- The numbering of sections: Please check the numbering of sections carefully; in the current paper, section 2 is numbered as section 1.
Response: Thanks, I have revised it.
- Section 2.1.: Section 2.1. on the second page starts with the examples on comparatives that make use of the dative-accusative marker xa/a.A (very) brief explanation of the functions and origins of this marker would be helpful for the reader, and the Section 2.1 could include a cross-reference to the Section 3 where the various functions of xa/a are explained in more detail.
Response: Thanks. In Section 2.1, an introduction to xa has been added.
- Section 2.2.: Section 2.2. contains a very interesting discussion on comparatives that make use of the element pi, 'to compare'. It would be good to add an example of Standard Mandarinto this section to illustrate more clearly that the form pi in Zhoutun is a cognate to Standard Mandarin, as well as demonstrate the point made in the section that examples (7b), (8) and (9) share some important properties with Standard Mandarin. (This may be obvious for readers who know Mandarin Chinese, but the paper may also be of interest for the readers who are not knowledgeable about Mandarin Chinese grammar).
Response: Thanks. The examples of pi comparatives in Mandarin has been listed in (3) and (11) in Section 2.
Section 2.2., second chapter: Mandarin Chinese's pi comparatives > Mandarin Chinese pi comparatives
Response: Thanks!
- The table 1: The formatting of Table 1 looks somewhat incomplete. Why is the column 'Form' let empty? In addition to type and form of the comparative constructions, it would be helpful for the readers to add a brief summary of origins of each construction.
Response: Thanks. The completeness of Table 1 has been augmented. As for the origin of each construction, it is not listed in Table 1 as it is discussed in Section 3.
- Conclusions: In the conclusions, it would be good to discuss the Zhoutun data more explicitly in relation to typological studies on comparatives. Is the distinction between comparative subject and attributive subject, as well as concrete and abstract measurement lacking from previous typologies altogether?
Response: Thanks for the insightful suggestion, which I have incorporated in the conclusion section. To the best of my knowledge, there appears to be a lack of systematic studies on the aforementioned two variables in descriptions in other languages and typological research on comparatives. Of course, it cannot be excluded that such studies exist in the literature but were not encountered in my review.

Reviewer 2 Report
This is an interesting, descriptive paper on Zhoutun comparative constructions. I find the language data presented to be useful, and I believe that the data makes a clear contribution to our understanding of comparative constructions in languages from the Qinghai-Gansu region.
To this reader, it is not yet fully clear how the main theoretical argument of the article--the distinction between concrete and abstract measurement, and comparative subject and attributive measurement--contributes to typological studies of comparatives. To make this argument, the authors would need to include a literature review on comparatives in typological literature more broadly (not only in the Gan-Qing dialects, and beyond Liu 2012 which is currently cited in the Introduction). For example, the authors state on page 3 "In typological studies, the distinction between ComS and ArtS has gotten little attention" but they have not provided any evidence for this statement or provided background to the reader about the existing typological literature on comparative constructions. Traditional works like Shopen (1985) Language Typology and Syntactic Description might be useful in this regard.
This reader found the argument that "the different position of ConM and AbsM in Zhoutun reflects the hybrid feature of Chinese and Tibetan and represents and 'incomplete' transition from the former to the latter" (page 3) to be very interesting and insightful. However, this argument is a bit buried and remains largely hypothetical in section 3. It would be helpful to focus more data and literature on this argument.
It would be helpful to include an explanation of the transcription process upfront.
Copy-editing needed for some sentences and for structural elements of the paper. For example, Section 2 is labeled as 1. Table 1 has a blank column under "Form." Direct quotations such as "remains unclear" (from Sandman and Simon) does not have a page number following the reference.
Oftentimes, there were paragraphs of 1-2 sentences after presenting a set of data that left the reader hanging and were hypothetical. For example, section 2.2 concludes with a statement that pi-Can is becoming more common but there is no evidence presented to support this statement. It would be helpful to use topic and concluding sentences in paragraphs, in order to fully frame and explain the relevance of each data point for the reader. Aim to explain the significance upfront, and then use the presented data to demonstrate the significance.
Author Response
Response to reviewer 2:
This is an interesting, descriptive paper on Zhoutun comparative constructions. I find the language data presented to be useful, and I believe that the data makes a clear contribution to our understanding of comparative constructions in languages from the Qinghai-Gansu region.
To this reader, it is not yet fully clear how the main theoretical argument of the article--the distinction between concrete and abstract measurement, and comparative subject and attributive measurement--contributes to typological studies of comparatives. To make this argument, the authors would need to include a literature review on comparatives in typological literature more broadly (not only in the Gan-Qing dialects, and beyond Liu 2012 which is currently cited in the Introduction). For example, the authors state on page 3 "In typological studies, the distinction between ComS and ArtS has gotten little attention" but they have not provided any evidence for this statement or provided background to the reader about the existing typological literature on comparative constructions. Traditional works like Shopen (1985) Language Typology and Syntactic Description might be useful in this regard.
Response: Thanks. I have conducted a further literature review on the typological study of comparative constructions, and have added references to Dixon (2012) and Stolz (2013) to supplement the existing sources.
This reader found the argument that "the different position of ConM and AbsM in Zhoutun reflects the hybrid feature of Chinese and Tibetan and represents and 'incomplete' transition from the former to the latter" (page 3) to be very interesting and insightful. However, this argument is a bit buried and remains largely hypothetical in section 3. It would be helpful to focus more data and literature on this argument.
Response: Thanks. In response to this issue, I have added a discussion in Section 3. I have parallely compared the comparative constructions in Mandarin Chinese and Tibetan, and have visually contrasted the comparative construction in Zhoutun with the aforementioned two, resulting in the conclusions presented above. I believe the revised presentation has improved credibility.
It would be helpful to include an explanation of the transcription process upfront.
Response: Thanks. In Section 1, I have added an explanation of the background of Zhoutun and the methodology for collecting the data.
Copy-editing needed for some sentences and for structural elements of the paper. For example, Section 2 is labeled as 1. Table 1 has a blank column under "Form." Direct quotations such as "remains unclear" (from Sandman and Simon) does not have a page number following the reference.
Response: Thanks. These issues have been addressed.
Oftentimes, there were paragraphs of 1-2 sentences after presenting a set of data that left the reader hanging and were hypothetical. For example, section 2.2 concludes with a statement that pi-Can is becoming more common but there is no evidence presented to support this statement. It would be helpful to use topic and concluding sentences in paragraphs, in order to fully frame and explain the relevance of each data point for the reader. Aim to explain the significance upfront, and then use the presented data to demonstrate the significance.
Response: Thank you for pointing out this issue. In the revised version, I have made corrections to the clarity and coherence of the article, striving to make the argument more robust.

Reviewer 3 Report
The manuscript deals with a topic which is relevant for both general linguistics and Chinese linguistics research tradition. The contents, referencing and readability show its high quality and originality.
Several strong points can be mentioned.
1. The study explores a fine-grained language-internal variation in the comparative constructions by also explaining their origins, development paths and cross-category relations. The variants observed in Zhoutun alone can be used to describe the typology of comparatives even in the entire North(east) Asia areas. For instance, the directional case comparative is observed in core Altaic languages plus Japanese, and the look-comparative is observed in Korean. This can be optionally mentioned too to show the typological relevance of Zhoutun from the areal perspective.
2. The study convincingly frames the explanation in a given contact scenario with Tibetic, crucially without ignoring the priority of internal reconstruction over contact explanation when dealing with development of a given language feature (see also Thomason 2010).
3. The study adds a more subtle categorical differentiation to the existing typological classification of comparatives, which is very useful for future scholarship and improvement of crosslinguistic comparative methods. Especially the newly discovered distinction between concrete vs. abstact measurement, which only show asymmetrical relation in Chinese varnaculars, has not received enough attention among typologists before the publication of the current study.
Several points can be considered for the improvement of the argumentation.
1. The topic prominence effect on the comparative constructions discussed in Section 2.4 does not only concern the 'and'-Cxn, but it can be used more vividly to explain also the differentiation between comparative subject and attributive subject, the former of which is pragmatically topic in the first three types (Sectinos 2.1 to 2.3), while the latter is at best a secondary topic. Parallels can be found, if needed, also from other neighbouring languages of Chinese in the east, for instance, Japanese, Korean and Manchu where double nominative construction is widely used for expressing external possession, such as the topical referent's age, property, etc. This will highlight the typological contribution of the current study even more and its relevance for a general discussion of the comparative constructions with a cross-category analysis with topichood, subjecthood and possession.
2. The argument against Altaic for Tibetic influence in Section 3 with concrete examples from contact languages and thorough explanation on the structural comparability is convincing. It can further be used to discuss with or argue against some previous studies which maintain that the Altaic layer of contact influence was deeper in the history whereas the Tibetic layer is more recent, with the Altaic one being more significant and responsible for the restructuring of northwestern Chinese typology (see e.g. Janhunen 2007, 2012).
3. Given that Zhoutun had changed its basic word order into SOV, the original “abnormal” word order of pi comparatives in Chinese became “normal” in Zhoutun pi-Cxn. - This statement is a good observation and claim. It can be even more strengthened by adding observation from Archaic Chinese in which the postadjectival standard was attested long earlier in yu- and guo-comparatives before the grammaticalisation of the verb pi 'to compare' (see Sun 1996: 10–11, 39).
4. However, as ‘look’ comparatives are not found in any Tibetan language outside of the Gan-Qing linguistic area, Sandman and Simon believe that ‘look’ comparatives are an independent development of the Gan-Qing linguistic area and their source “remains unclear”. - This is another interesting point, and it might be useful to compare with Korean which predominantly use look-comparative and is independent of this Gan-Qing contact zone.
Several issues in the manuscript structure can also be considered for better readability.
1. The sources and gathering of the data can be explained more explicitly, even though it is clear without saying from the manuscript that it is the author's own collected examples.
2. The introduction of the four key elements: ComS, AttS, AbsM, ConM, comes in the mix with the presentation of xa-comparative. It might be easier for readers if these concepts are given under the main section header 2 before starting to go into different types in the subsequent subsections.
3. Check the numberings and their format to be consistent throughout the manuscript.
4. In the renowned 那场火幸亏消防队来得快 nei chang huo xingkui xiaofangdui lai de kuai that cl fire fortunate fire.brigade come comp fast ‘That fire, fortunately the fire brigade came quickly’, for instance, nei chang huo ‘that fire’ is not an argument of the predicate lai de kuai ‘come quickly’. - This example before example (27) should also be presented as a numbered and glossed example. As it is now, readers without Chinese knowledge can easily get lost in the text.
5. Example (28) has translation swapped between the two clauses with 'less' and 'more', which should infact be in the other way around(?).
References
Janhunen, Juha. 2007. Typological interaction in the Qinghai linguistic complex. Studia Orientalia 101. 85–102.
Janhunen, Juha. 2012. On the hierarchy of structural convergence in the Amdo Sprachbund. In P. Suihkonen, C. Bernard & V. Solovyev (eds.), Argument structure and grammatical relations: A cross-linguistic typology, 177–189. Amsterdam: John Benjamins.
Thomason, Sarah Grey. 2010. Contact Explanations in Linguistics. In Raymond Hickey (ed.), The Handbook of Language Contact, 31–47. Oxford: Blackwell Publishing.
Author Response
Response to reviewer 3:
The manuscript deals with a topic which is relevant for both general linguistics and Chinese linguistics research tradition. The contents, referencing and readability show its high quality and originality.
Several strong points can be mentioned.
- The study explores a fine-grained language-internal variation in the comparative constructions by also explaining their origins, development paths and cross-category relations. The variants observed in Zhoutun alone can be used to describe the typology of comparatives even in the entire North(east) Asia areas. For instance, the directional case comparative is observed in core Altaic languages plus Japanese, and the look-comparative is observed in Korean. This can be optionally mentioned too to show the typological relevance of Zhoutun from the areal perspective.
Response: The reviewer is thanked for bringing to light relevant information. It is indeed the case that the comparative markers in Altaic languages are predominantly syncretism with the ablative markers, as discussed in Zhou (2020a). The situation in Japanese and Korean was previously not known to me, and I am grateful to the reviewer for providing this information and enabling me to gain a deeper understanding of the relevant phenomena. The revised draft includes mention of the situation in Japanese and Korean.
- The study convincingly frames the explanation in a given contact scenario with Tibetic, crucially without ignoring the priority of internal reconstruction over contact explanation when dealing with development of a given language feature (see also Thomason 2010).
- The study adds a more subtle categorical differentiation to the existing typological classification of comparatives, which is very useful for future scholarship and improvement of crosslinguistic comparative methods. Especially the newly discovered distinction between concrete vs. abstact measurement, which only show asymmetrical relation in Chinese varnaculars, has not received enough attention among typologists before the publication of the current study.
Response to 2 and 3: Thanks very much!
Several points can be considered for the improvement of the argumentation.
- The topic prominence effect on the comparative constructions discussed in Section 2.4 does not only concern the 'and'-Cxn, but it can be used more vividly to explain also the differentiation between comparative subject and attributive subject, the former of which is pragmatically topic in the first three types (Sectinos 2.1 to 2.3), while the latter is at best a secondary topic. Parallels can be found, if needed, also from other neighbouring languages of Chinese in the east, for instance, Japanese, Korean and Manchu where double nominative construction is widely used for expressing external possession, such as the topical referent's age, property, etc. This will highlight the typological contribution of the current study even more and its relevance for a general discussion of the comparative constructions with a cross-category analysis with topichood, subjecthood and possession.
Response: I am grateful for the insightful comments made by the reviewer. I concur with the reviewer's assessment that the distinction between ComS and AttS reflects a topic-prominent characteristic. Liu (2012) provides a detailed discussion on this point. I have briefly mentioned this issue in the revised conclusion. The reviewer's observation regarding the phenomenon in Japanese, Korean, and Manchu is also highly enlightening. Due to limitations of my capacity, I have yet to gain a deeper understanding of this issue, but I believe it can be included in future research efforts.
- The argument against Altaic for Tibetic influence in Section 3 with concrete examples from contact languages and thorough explanation on the structural comparability is convincing. It can further be used to discuss with or argue against some previous studies which maintain that the Altaic layer of contact influence was deeper in the history whereas the Tibetic layer is more recent, with the Altaic one being more significant and responsible for the restructuring of northwestern Chinese typology (see e.g. Janhunen 2007, 2012).
Response: Yes, the view held in a series of studies I conducted is that the layer of influence of the Tibetan language on the Gansu-Qinghai linguistic area was earlier, while that of the Altaic languages was relatively recent.
- Given that Zhoutun had changed its basic word order into SOV, the original “abnormal” word order of pi comparatives in Chinese became “normal” in Zhoutun pi-Cxn. - This statement is a good observation and claim. It can be even more strengthened by adding observation from Archaic Chinese in which the postadjectival standard was attested long earlier in yu- and guo-comparatives before the grammaticalisation of the verb pi 'to compare' (see Sun 1996: 10–11, 39).
Response: Thanks. Indeed, the yu- and guo-comparatives are in harmony with the VO word order. This is addressed in the revised version in Footnote 7.
4.However, as ‘look’ comparatives are not found in any Tibetan language outside of the Gan-Qing linguistic area, Sandman and Simon believe that ‘look’ comparatives are an independent development of the Gan-Qing linguistic area and their source “remains unclear”. - This is another interesting point, and it might be useful to compare with Korean which predominantly use look-comparative and is independent of this Gan-Qing contact zone.
Response: Thanks for bringing this to my attention. In the revised version, I have acknowledged the source of the comparative marker in Korean, which originated from "look," by referencing Rhee (2021).
Several issues in the manuscript structure can also be considered for better readability.
- The sources and gathering of the data can be explained more explicitly, even though it is clear without saying from the manuscript that it is the author's own collected examples.
- The introduction of the four key elements: ComS, AttS, AbsM, ConM, comes in the mix with the presentation of xa-comparative. It might be easier for readers if these concepts are given under the main section header 2 before starting to go into different types in the subsequent subsections.
- Check the numberings and their format to be consistent throughout the manuscript.
- In the renowned 那场火幸亏消防队来得快 nei chang huo xingkui xiaofangdui lai de kuai that cl fire fortunate fire.brigade come comp fast ‘That fire, fortunately the fire brigade came quickly’, for instance, nei chang huo ‘that fire’ is not an argument of the predicate lai de kuai ‘come quickly’. - This example before example (27) should also bepresented as a numbered and glossed example. As it is now, readers without Chinese knowledge can easily get lost in the text.
5.Example (28) has translation swapped between the two clauses with 'less' and 'more', which should in fact be in the other way around(?).
Response to 1-5: Thank you for the reviewer for pointing out these issues, I have made all revisions.
References
Janhunen, Juha. 2007. Typological interaction in the Qinghai linguistic complex. Studia Orientalia 101. 85–102.
Janhunen, Juha. 2012. On the hierarchy of structural convergence in the Amdo Sprachbund. In P. Suihkonen, C. Bernard & V. Solovyev (eds.), Argument structure and grammatical relations: A cross-linguistic typology, 177–189. Amsterdam: John Benjamins.
Thomason, Sarah Grey. 2010. Contact Explanations in Linguistics. In Raymond Hickey (ed.), The Handbook of Language Contact, 31–47. Oxford: Blackwell Publishing.

Round 2
Reviewer 2 Report
This reviewer found that the authors' edits have substantially improved the quality of argumentation, and the contextualization of their theoretical contribution. While references remain a bit vague (for example, "see, among others, Dixon 2012" rather than providing detailed explanations of the connections to existing literature), the main argument is now clear and substantiated. This reviewer still recommends some more robust engagement with these sources through quotation and detailed citation, but it is entirely not necessary as the main contribution of the article is now clear on the basis of the presented data.
I found that the details provided about Zhoutun Village and the methods of fieldwork, included in the Introduction, strengthened the contribution of the article. The authors may consider moving the last paragraph of section 1 ("The data pertaining to Zhoutun presented in this study was acquired...") near the beginning of the Introduction, so that the Introduction ends with a clear statement of the theoretical contribution o the article.
The authors have successfully used linking phrases in section 2 to substantiate their claims in relation to a broader corpus, and to explain to the reader exactly how the presented data provides evidence of their claims.
The Conclusion is now clear and reiterates the contributions demonstrated throughout the article.
The major errors that required copy-editing, such as the labeling of Section 2 and Table 1, have now been resolved. This reviewer recommends minor copy-editing for ease of reading.